# Rapid SINS Two-Position Ground Alignment Scheme Based on Piecewise Combined Kalman Filter and Azimuth Constraint Information

**DOI:** 10.3390/s19051125

**Published:** 2019-03-05

**Authors:** Lu Zhang, Wenqi Wu, Maosong Wang

**Affiliations:** College of Intelligence Science and Engineering, National University of Defense Technology, Changsha 410073, China; wangmaosong12@hotmail.com

**Keywords:** piecewise combined Kalman filter, SINS, relative azimuth constraints, two-position initial alignment

## Abstract

The accuracy and rate of convergence are two important performance factors for initial ground alignment of a strapdown inertial navigation system (SINS). For navigation-grade SINS, gyro biases and accelerometer offsets can be modeled as constant values during the alignment period, and they can be calibrated through two-position ground alignment schemes. In many situations for SINS ground alignment, the azimuth of the vehicle remains nearly constant. This quasi-stationary alignment information can be used as an augmented measurement. In this paper, a piecewise combined Kalman filter utilizing relative azimuth constraint (RATP) is proposed to improve the alignment precision and to reduce the time consumption for error convergence. It is presented that a piecewise time-invariant linear system can be combined into a whole extended time-invariant linear system so that a piecewise combined Kalman filter can be designed for state estimation. A two-position ground alignment algorithm for SINS is designed based on the proposed piecewise combined Kalman filter. Numerical simulations and experimental results show its superiority to the conventional algorithms in terms of accuracy and the rate of convergence.

## 1. Introduction

Ground alignment is divided into integrated alignment and self-alignment. The initial-self alignment process generally consists of two stages called coarse alignment and fine alignment [1,2,3,4,5,6]. This paper focuses on fine alignment, which plays a pivotal role in the alignment accuracy and the rate of convergence. The rotation modulation technique (RMT) is one of the effective techniques available for self-alignment without any external information [7,8], and improves the alignment performance compared with traditional alignment schemes [9,10]. The RMT includes single-axis modulation, dual-axis modulation, and tri-axis modulation [11]. Single-axis modulation is much more widely used in practice because it is simpler and easier to operate [1,9,12]. In recent years, scholars have conducted much research on single-axis modulation. Acharya et al. [13] used augmented measurement to improve the rate of convergence of azimuth error. Refs. [14,15,16] utilized adding angular rates as measurements to provide good alignment performance, while Ref. [17] proposed an improved initial alignment based on horizontal alignment information in inertial frame, and the authors of [18] employed linear equality to improve the observability of strapdown inertial navigation system (SINS). A self-alignment method was put forward based on three vectors of gravitational apparent motion in an inertial frame [19]. Ref. [20] proposed a dual mathematical calculation system (DMCS) to improve the alignment accuracy when there is a large initial misalignment angle. Furthermore, nonlinear filters [12,21] were also used to solve the large misalignment angle, but system model and computation became complex. Meanwhile, the error of coarse alignment is less than one degree in general situations [2,6], therefore the Kalman filter is mostly employed, although an alternative method was proposed to perform the initial alignment [22]. Hence, we consider the initial misalignment angle as a small angle after coarse alignment in this paper.

Different schemes can be implemented for single-axis rotation, such as two-position alignment schemes [13,15,16,23,24,25,26,27], reciprocating rotation schemes, continuously rotation schemes [7,9,18], and others. Two-position alignment schemes can improve the observability of the system and allow nearly all error states to be estimated, including gyro biases and accelerometer offsets. The heading change of the IMU (inertial measurement unit) is generally arranged as approximately 180° for two-position alignment with the same time duration to achieve optimal estimation results [23,25,26].

The north and east velocities are generally used as measurements in the Kalman filter for SINS self-initial alignment [23,28]. Recently, adding angular rates as measurements to get faster convergence was proposed [13,15,16,17,21]. The observation models are established in the body frame [13,17] and navigation frame [15,16,21], respectively.

In many inertial navigation applications, both high initial alignment precision and fast error convergence are required [15,16,18]. Ref. [16] considered the influence of cross-correlation between system noises and measurement noises. The output of the equivalent east gyro in geographic frame [29] and a nonlinear state constraint [18] are used to improve the speed of convergence. It is shown from the above literature that adding effective measurement information can accelerate the convergence speed of azimuth error. The current two-position alignment research is mainly concentrated on adding gyro information, and the vehicle rotation rate constraint as measurements [13,14,15,16,17,18,21,30]. The performance of this method is limited in practical applications because the observation noise related with gyro information and the vehicle rotation rate constraint can be generally much larger, even under small angular vibrations.

On the other hand, for ground self-alignment applications, the SINS is approximately stationary with respect to Earth, and the azimuth remained nearly constant during the quasi-stationary alignment period. The azimuth constraint information could be used as observation, based on which the vehicle velocity, attitude, and IMU errors could be estimated and calibrated. The observation noise related with the IMU azimuth constraint information is generally much smaller under quasi-stationary situations.

Using the above considerations, a piecewise combined Kalman filter is proposed in this paper to improve the accuracy and rate of convergence of the two-position alignment based on IMU azimuth constraint information.

The rest of the paper is organized as follows. The piecewise combine Kalman filter, the new two-position alignment models and alignment performance by simulation are shown in Section 2. Results and discussions of experiments are presented in Section 3. Conclusions are given in Section 4.

## 2. Materials and Methods

The concept of the piecewise combined time-invariant linear system model and the piecewise combined Kalman filter is proposed in this section. The conventional two-position alignment model as well as the augmented observation model based on gyro information and vehicle rotation rate constraint are briefly introduced. Finally, an improved two-position alignment scheme is presented based on the piecewise combined Kalman filter, which utilizes IMU azimuth constraint information for self-alignment.

### 2.1. The Piecewise Combined Time-Invariant Linear System Model

The system equation and observation equation of a piecewise time-invariant linear system [31,32] can be expressed as follows:(1)$$\left[\begin{array}{c} {\dot{x}}_{1}\left(t\right)\\ {\dot{x}}_{2} \end{array}\right]=\left[\begin{array}{cc} F\left(t\right) & G\left(t\right)\\ 0 & 0 \end{array}\right]\left[\begin{array}{c} x_{1}\left(t\right)\\ x_{2} \end{array}\right]+\left[\begin{array}{c} w\left(t\right)\\ 0 \end{array}\right],$$
(2)$$z=\left[\begin{array}{cc} H & 0 \end{array}\right]\left[\begin{array}{c} x_{1}\left(t\right)\\ x_{2} \end{array}\right]+\upsilon \left(t\right),$$
where matrix F(t) and G(t) satisfy the following conditions while T is the system running time:(3)$$F\left(t\right)=\{\begin{array}{c} F_{1}\;t\in \left(0,\frac{T}{2}\right]\\ F_{2}\;t\in \left(\frac{T}{2},T\right] \end{array}\;G\left(t\right)=\{\begin{array}{c} G_{1}\;t\in \left(0,\frac{T}{2}\right]\\ G_{2}\;t\in \left(\frac{T}{2},T\right] \end{array}.$$

Therefore, the original piecewise time-invariant linear system model shown in Equation (1) can be rewritten as the following piecewise combined system model:(4)$$\left[\begin{array}{c} {\dot{x}}_{11}\left(t_{1}\right)\\ {\dot{x}}_{12}\left(t_{2}\right)\\ {\dot{x}}_{2} \end{array}\right]=\left[\begin{array}{ccc} F_{1} & 0 & G_{1}\\ 0 & F_{2} & G_{2}\\ 0 & 0 & 0 \end{array}\right]\left[\begin{array}{c} x_{11}\left(t_{1}\right)\\ x_{12}\left(t_{2}\right)\\ x_{2} \end{array}\right]+\left[\begin{array}{c} w_{1}\left(t_{1}\right)\\ w_{2}\left(t_{2}\right)\\ 0 \end{array}\right],$$
where $t_{1}\in \left(0,\frac{T}{2}\right]$, $t_{2}\in \left(\frac{T}{2},T\right]$.

Meanwhile, there are observations associated with two state vectors $x_{11}\left(t_{1}\right)$ and $x_{12}\left(t_{2}\right)$. Therefore, the observation model shown in Equation (2) can be further expressed as

(5)$$\bar{z}=\left[\begin{array}{ccc} H & 0 & 0\\ 0 & H & 0\\ H_{1} & H_{2} & 0 \end{array}\right]\left[\begin{array}{c} x_{11}\left(t_{1}\right)\\ x_{12}\left(t_{2}\right)\\ x_{2} \end{array}\right]+\left[\begin{array}{c} \upsilon_{1}\left(t_{1}\right)\\ \upsilon_{2}\left(t_{2}\right)\\ \upsilon_{r}\left(t_{12}\right) \end{array}\right].$$

Consequently, a piecewise combined Kalman filter based on the system dynamic model in Equation (4) and observation model in Equation (5) can be constructed, which is named as PC-KF in the following sections. The PC-KF can be widely used in piecewise constant linear systems for state estimation, while we only focus its application on two-position alignment schemes in this paper.

### 2.2. The Conventional SINS Two-Position Alignment Model

The conventional error state model for SINS rapid self-alignment under quasi-stationary conditions, which is similar to Equation (1), can be written as follows [16,18,28]:(6)$$\begin{array}{c} \left[\begin{array}{c} {\dot{x}}_{1}\left(t\right)\\ {\dot{x}}_{2} \end{array}\right]=\left[\begin{array}{cc} F & G\left(t\right)\\ 0_{4\times 5} & 0_{4\times 4} \end{array}\right]\left[\begin{array}{c} x_{1}\left(t\right)\\ x_{2} \end{array}\right]+\left[\begin{array}{c} w\left(t\right)\\ 0_{1\times 4} \end{array}\right]\;\\ =Ax\left(t\right)+w\left(t\right),\;w\left(t\right)∼N\left(0,Q\right), \end{array}$$
where $x_{1}\left(t\right)={\left[\begin{array}{ccccc} \phi_{N}^{} & \phi_{E}^{} & \phi_{D}^{} & \delta v_{N} & \delta v_{E} \end{array}\right]}^{T}$ and $x_{2}={\left[\begin{array}{cccc} \varepsilon_{x}^{b} & \varepsilon_{y}^{b} & \nabla_{x}^{b} & \nabla_{y}^{b} \end{array}\right]}^{T}$; $\phi^{n}={\left[\begin{array}{ccc} \phi_{N}^{} & \phi_{E}^{} & \phi_{D}^{} \end{array}\right]}^{T}$ represents the attitude error vector; $\delta v^{n}={\left[\begin{array}{ccc} \delta v_{N} & \delta v_{E} & \delta v_{D} \end{array}\right]}^{T}$ represents the velocity error vector expressed in local navigation frame n. $\varepsilon^{b}={\left[\begin{array}{ccc} \varepsilon_{x}^{b} & \varepsilon_{y}^{b} & \varepsilon_{z}^{b} \end{array}\right]}^{T}$ and $\nabla^{b}={\left[\begin{array}{ccc} \nabla_{x}^{b} & \nabla_{y}^{b} & \nabla_{z}^{b} \end{array}\right]}^{T}$ represent the gyro triad bias vector and the accelerometer triad offset vector, respectively, which can be seen as constant values for rapid SINS self-alignment. $\delta v_{D}$ and $\nabla_{z}^{b}$ are generally not considered for SINS self-alignment in land vehicle applications, in addition, $\varepsilon_{z}^{b}$ is difficult to estimate accurately in a short time, so it is not considered. Components of w(t) are process noises of $\phi^{n}$ and $\delta v^{n}$ which can be denoted in a combined vector as $w\left(t\right)={\left[\begin{array}{ccccc} w_{\phi N} & w_{\phi E} & w_{\phi D} & w_{vN} & w_{vE} \end{array}\right]}^{T}$. The process covariance matrix of w(t) is Q which can be written as:(7)$$Q=\left[\begin{array}{cc} n_{g}^{2}I_{3\times 3} & 0_{3\times 2}\\ 0_{2\times 3} & n_{a}^{2}I_{2\times 2} \end{array}\right]$$

The matrix F takes the following form:(8)$$F=\left[\begin{array}{ccccc} 0 & \omega_{ieD}^{n} & 0 & 0 & 0\\ -\omega_{ieD}^{n} & 0 & \omega_{ieN}^{n} & 0 & 0\\ 0 & -\omega_{ieN}^{n} & 0 & 0 & 0\\ 0 & g & 0 & 0 & 2\omega_{ieD}^{n}\\ -g & 0 & 0 & -2\omega_{ieD}^{n} & 0 \end{array}\right].$$

Similarly, we have
(9)$$G\left(t\right)=\left[\begin{array}{cccc} -C_{11} & -C_{12} & 0 & 0\\ -C_{21} & -C_{22} & 0 & 0\\ -C_{31} & -C_{32} & 0 & 0\\ 0 & 0 & C_{11} & C_{12}\\ 0 & 0 & C_{21} & C_{22} \end{array}\right].$$
$C_{ij}\left(i,j=1,2,3\right)$ represents the element of $C_{b}^{n}$ which denotes the attitude transformation matrix from b frame to n frame. $\omega_{ie}^{n}={\left[\begin{array}{ccc} \omega_{ieN}^{n} & \omega_{ieE}^{n} & \omega_{ieD}^{n} \end{array}\right]}^{T}$ is the earth rotation rate with respect to the inertial frame expressed in n frame, while $\omega_{en}^{n}$ is the rotation rate of the n frame with respect to the e frame expressed in n frame. f represents the specific force acceleration. For quasi-stationary base initial alignment, the following simplification can be made: $f^{n}=-g^{n}$, $\omega_{en}^{n}=0$.

The conventional north and east velocity error observation model is given by Refs. [10,11,12,13]
(10)$$\begin{array}{c} z_{1}\left(t\right)={\left[\begin{array}{cc} \delta v_{N} & \delta v_{E} \end{array}\right]}^{T}=\left[\begin{array}{ccc} 0_{2\times 3} & I_{2\times 2} & 0_{2\times 4} \end{array}\right]x\left(t\right)+\upsilon_{1}\left(t\right)\\ =H_{1}\left(t\right)x\left(t\right)+\upsilon_{1}\left(t\right),\;\upsilon_{1}\left(t\right)∼N\left(0,R_{1}\right) \end{array}$$
where $\upsilon_{1}\left(t\right)$ represents the observation noise vector; $R_{1}$ represents the observation noise covariance matrix.

(11)$$R_{1}=\left[\begin{array}{cc} r_{v}^{2} & 0\\ 0 & r_{v}^{2} \end{array}\right]$$

The conventional Kalman filter KF) can be designed based on the observation model shown in Equation (10) and the dynamic error model shown in Equation (6). The corresponding conventional two-position initial alignment algorithm is noted as TP in the following sections.

### 2.3. The Conventional Augmented Observation Model Based on Angular Rate Measurements

As shown in Refs. [13,15,16], angular rate measurements and the vehicle rotation rate constraint can be used in order to improve the convergence rate of alignment. Theoretically, $\omega_{ie}^{n}$ can be written as
(12)$$\omega_{ie}^{n}=C_{b}^{n}\omega_{ib}^{b}-\omega_{nb}^{n}-\omega_{en}^{n}.$$
However, during the ground initial alignment period, $\omega_{nb}^{n}$ and $\omega_{en}^{n}$ can be seen as nearly zero if the vehicle or the SINS is in quasi-stationary situations. The estimated transformation matrix ${\tilde{C}}_{b}^{n}$ is used for attitude update and ${\bar{C}}_{b}^{n}=\left[I-\phi^{n}\times \right]C_{b}^{n}$. Therefore,
(13)$$\begin{array}{l} C_{n}^{b}\left(C_{b}^{n}\omega_{ib}^{b}-\omega_{nb}^{n}-\omega_{en}^{n}\right)\approx \left({\tilde{\omega }}_{ib}^{b}-\varepsilon^{b}\right)\\ \Rightarrow {\tilde{C}}_{n}^{b}\left(I-\left[\phi^{n}\times \right]\right)\omega_{ie}^{n}=\left({\tilde{\omega }}_{ib}^{b}-\varepsilon^{b}\right)\\ \Rightarrow \left(I-\left[\phi^{n}\times \right]\right)\omega_{ie}^{n}={\tilde{C}}_{b}^{n}\left({\tilde{\omega }}_{ib}^{b}-\varepsilon^{b}\right)\\ \Rightarrow {\tilde{C}}_{b}^{n}{\tilde{\omega }}_{ib}^{b}=\left(I-\left[\phi^{n}\times \right]\right)\omega_{ie}^{n}+{\tilde{C}}_{b}^{n}\varepsilon^{b}\\ \Rightarrow {\tilde{C}}_{b}^{n}{\tilde{\omega }}_{ib}^{b}-\omega_{ie}^{n}=\left[\omega_{ie}^{n}\times \right]\phi^{n}+{\tilde{C}}_{b}^{n}\varepsilon^{b} \end{array}$$

If we ignore small quantities and take the equivalent east gyro error as measurement, then the following equation can be obtained:(14)$$\left[\begin{array}{ccc} 0 & 1 & 0 \end{array}\right]\left({\tilde{C}}_{b}^{n}{\tilde{\omega }}_{ib}^{b}-\omega_{ie}^{n}\right)=\delta \omega_{ieE}^{n}=-\omega_{ieD}^{n}\phi_{N}+\omega_{ieN}^{n}\phi_{D}+C_{21}\varepsilon_{x}^{b}+C_{22}\varepsilon_{y}^{b}$$

As can be seen in Equation (14), the augmented observation model can be obtained by adding angular rate measurements. Then, the observation model can be expressed as
(15)$$\begin{array}{ll} z_{2}\left(t\right) & =\left[\begin{array}{c} z_{1}\left(t\right)\\ \delta \omega_{ieE}^{n} \end{array}\right]=\left[\begin{array}{ccccccc} 0_{2\times 1} & 0_{2\times 1} & 0_{2\times 1} & I_{2\times 2} & 0_{2\times 1} & 0_{2\times 1} & 0_{2\times 2}\\ -\omega_{ieD}^{n} & 0 & \omega_{ieN}^{n} & 0_{1\times 2} & C_{21} & C_{22} & 0_{1\times 2} \end{array}\right]x\left(t\right)+\left[\begin{array}{c} \upsilon_{1}\left(t\right)\\ \upsilon_{\omega E}\left(t\right) \end{array}\right]\\ & =H_{2}\left(t\right)x\left(t\right)+\upsilon_{2}\left(t\right),\;\upsilon_{2}\left(t\right)∼N\left(0,R_{2}\right) \end{array}$$
where $R_{2}$ takes the following form:(16)$$R_{2}=\left[\begin{array}{cc} R_{1} & 0_{2\times 1}\\ 0_{1\times 2} & r_{\omega E}^{2} \end{array}\right]$$
$\upsilon_{\omega E}\left(t\right)$ represents the angular rate measurements noise about the east axis, and
(17)$$\upsilon_{\omega E}\left(t\right)\approx \left[\begin{array}{ccc} 0 & 1 & 0 \end{array}\right]\left(\omega_{nb}^{n}+\omega_{en}^{n}\right)$$

Here, the corresponding angular rate measurement-augmented two-position initial alignment algorithm is denoted as ARTP. Although sometimes the angular vibration amplitude of the vehicle is very small, the vibration angular rate can be large, especially along horizontal directions. For example, if the vehicle engine is still working in the quasi-stationary situations, $\omega_{nb}^{n}$ and $\omega_{en}^{n}$ are no longer zero, which lead to the changing of measurement noise $\upsilon_{\omega E}\left(t\right)$. Thus, the ARTP algorithm has limitations in these cases.

### 2.4. The Piecewise Combined Kalman Filter for Improved Two-Position Initial Alignment

In this section, a piecewise combined Kalman filter (noted as PC-KF) is designed for the two-position SINS initial alignment to improve the azimuth accuracy and the rate of convergence.

For the two-position initial alignment, we suppose that the SINS is in state A when the IMU is in the first position, and in state B after the IMU has rotated 180° while the two positions have the same time duration [23].

Differently from the conventional SINS alignment system model shown in Equation (6), the combined system model expands to 14 dimensions containing both error states in state A and state B. The system model can be described according to Equation (4):(18)$$\begin{array}{ll} \dot{\bar{x}}\left(t\right) & =\left[\begin{array}{ccc} F & 0_{5\times 5} & G_{1}\\ 0_{5\times 5} & F & G_{2}\\ 0_{4\times 5} & 0_{4\times 5} & 0_{4\times 4} \end{array}\right]\bar{x}\left(t\right)+\left[\begin{array}{c} G_{1}w\left(t_{1}\right)\\ G_{2}w\left(t_{2}\right)\\ 0_{1\times 4} \end{array}\right]\;\\ & =\bar{A}\bar{x}\left(t\right)+\left[\begin{array}{ccc} G_{1} & 0_{5\times 5} & 0_{5\times 4}\\ 0_{5\times 5} & G_{2} & 0_{5\times 4}\\ 0_{4\times 5} & 0_{4\times 5} & 0_{4\times 4} \end{array}\right]\left[\begin{array}{c} \bar{w}\left(t\right)\\ 0_{1\times 4} \end{array}\right]\;\bar{w}\left(t\right)∼N\left(0,\bar{Q}\right) \end{array}$$
where $\bar{x}\left(t\right)={\left[\begin{array}{cccccccccccccc} \phi_{NA} & \phi_{EA} & \phi_{DA} & \delta v_{NA} & \delta v_{EA} & \phi_{NB} & \phi_{EB} & \phi_{DB} & \delta v_{NB} & \delta v_{EB} & \varepsilon_{x}^{b} & \varepsilon_{y}^{b} & \nabla_{x}^{b} & \nabla_{y}^{b} \end{array}\right]}^{T}$. The covariance matrix of $\bar{w}\left(t\right)$ is $\bar{Q}$, which is written as
(19)$$\bar{Q}=\left[\begin{array}{cc} Q & 0_{5\times 5}\\ 0_{5\times 5} & Q \end{array}\right]$$

The matrixes $G_{1}$ and $G_{2}$ take the following form:(20)$$G_{1}=\left[\begin{array}{cccc} -C_{11A} & -C_{12A} & 0 & 0\\ -C_{21A} & -C_{22A} & 0 & 0\\ -C_{31A} & -C_{32A} & 0 & 0\\ 0 & 0 & C_{11A} & C_{12A}\\ 0 & 0 & C_{21A} & C_{22A} \end{array}\right],G_{2}=\left[\begin{array}{cccc} -C_{11B} & -C_{12B} & 0 & 0\\ -C_{21B} & -C_{22B} & 0 & 0\\ -C_{31B} & -C_{32B} & 0 & 0\\ 0 & 0 & C_{11B} & C_{12B}\\ 0 & 0 & C_{21B} & C_{22B} \end{array}\right].$$
$C_{bA}^{n}$ and $C_{bB}^{n}$ represent the attitude transformation matrix in state A and state B. $C_{ijA}\left(i,j=1,2,3\right)$ and $C_{ijB}\left(i,j=1,2,3\right)$ represent the element of attitude matrix $C_{bA}^{n}$ and $C_{bB}^{n}$, respectively.

The observation model is constructed by taking the north velocity error, the east velocity error, and the relative azimuth error angle as the measurements. The relative azimuth error angle can be obtained based on the IMU azimuth relationship between state A and state B. ${\bar{C}}_{b_{A}}^{n}$ and ${\tilde{C}}_{b_{B}}^{n}$ are the computational values of $C_{b}^{n}$ in two states:(21)$${\tilde{C}}_{b_{A}}^{n}=\left[I-\phi_{A}^{n}\times \right]C_{b_{A}}^{n}\;{\tilde{C}}_{b_{B}}^{n}=\left[I-\phi_{B}^{n}\times \right]C_{b_{B}}^{n}$$
in which $\phi_{A}^{n}=\left[\begin{array}{ccc} \phi_{NA} & \phi_{EA} & \phi_{DA} \end{array}\right]$, $\phi_{B}^{n}=\left[\begin{array}{ccc} \phi_{NB} & \phi_{EB} & \phi_{DB} \end{array}\right]$. The relationship between ${\bar{C}}_{b_{A}}^{n}$ and ${\tilde{C}}_{b_{B}}^{n}$ can be written as
(22)$$\begin{array}{l} C_{b_{A}}^{b_{B}}={\left(C_{b_{B}}^{n}\right)}^{T}C_{b_{A}}^{n}={\left(\left[I+\phi_{B}\times \right]{\tilde{C}}_{b_{B}}^{n}\right)}^{T}\left[I+\phi_{A}\times \right]{\tilde{C}}_{b_{A}}^{n}\\ ={\tilde{C}}_{n}^{b_{B}}\left[I-\phi_{B}\times \right]\left[I+\phi_{A}\times \right]{\tilde{C}}_{b_{A}}^{n}\\ \approx {\tilde{C}}_{n}^{b_{B}}\left[I+\left(\phi_{A}-\phi_{B}\right)\times \right]{\tilde{C}}_{b_{A}}^{n}\\ ={\tilde{C}}_{b_{A}}^{b_{B}}+\left[\begin{array}{ccc} {\tilde{c}}_{11B} & {\tilde{c}}_{21B} & {\tilde{c}}_{31B}\\ {\tilde{c}}_{12B} & {\tilde{c}}_{22B} & {\tilde{c}}_{32B}\\ {\tilde{c}}_{13B} & {\tilde{c}}_{23B} & {\tilde{c}}_{33B} \end{array}\right]\left[\begin{array}{ccc} 0 & -\left(\phi_{DA}-\phi_{DB}\right) & \left(\phi_{EA}-\phi_{EB}\right)\\ \left(\phi_{DA}-\phi_{DB}\right) & 0 & -\left(\phi_{NA}-\phi_{NB}\right)\\ -\left(\phi_{EA}-\phi_{EB}\right) & \left(\phi_{NA}-\phi_{NB}\right) & 0 \end{array}\right]\left[\begin{array}{ccc} {\tilde{c}}_{11A} & {\tilde{c}}_{12A} & {\tilde{c}}_{13A}\\ {\tilde{c}}_{21A} & {\tilde{c}}_{22A} & {\tilde{c}}_{23A}\\ {\tilde{c}}_{31A} & {\tilde{c}}_{32A} & {\tilde{c}}_{33A} \end{array}\right] \end{array}$$

Suppose the horizontal Euler angles between state A and state B are $\Delta \gamma_{AB}$ and $\Delta \theta_{AB}$, while the azimuth change is $\pi +\Delta \varphi_{AB}$. $\Delta \gamma_{AB}$, $\Delta \theta_{AB}$ and $\Delta \varphi_{AB}$ are seen as unknown small angles. Thus,
(23)$$\begin{array}{ll} C_{b_{A}}^{b_{B}} & \approx \left[\begin{array}{ccc} 1 & 0 & \Delta \theta_{AB}\\ 0 & 1 & -\Delta \gamma_{AB}\\ -\Delta \theta_{AB} & \Delta \gamma_{AB} & 1 \end{array}\right]\left[\begin{array}{ccc} \cos\left(\pi +\Delta \varphi_{AB}\right) & \sin\left(\pi +\Delta \varphi_{AB}\right) & 0\\ -\sin\left(\pi +\Delta \varphi_{AB}\right) & \cos\left(\pi +\Delta \varphi_{AB}\right) & 0\\ 0 & 0 & 1 \end{array}\right]\\ & =\left[\begin{array}{ccc} \cos\left(\pi +\Delta \varphi_{AB}\right) & \sin\left(\pi +\Delta \varphi_{AB}\right) & \Delta \theta_{AB}\\ -\sin\left(\pi +\Delta \varphi_{AB}\right) & \cos\left(\pi +\Delta \varphi_{AB}\right) & -\Delta \gamma_{AB}\\ -\Delta \theta_{AB}\cos\left(\pi +\Delta \varphi_{AB}\right)-\Delta \gamma_{AB}\sin\left(\pi +\Delta \varphi_{AB}\right) & -\Delta \theta_{AB}\sin\left(\pi +\Delta \varphi_{AB}\right)+\Delta \gamma_{AB}\cos\left(\pi +\Delta \varphi_{AB}\right) & 1 \end{array}\right]\\ & \approx \left[\begin{array}{ccc} -1 & -\Delta \varphi_{AB} & \Delta \theta_{AB}\\ \Delta \varphi_{AB} & -1 & -\Delta \gamma_{AB}\\ \Delta \theta_{AB} & -\Delta \gamma_{AB} & 1 \end{array}\right] \end{array}$$

${\tilde{C}}_{b_{A}}^{b_{B}}$ can be rewritten as
(24)$${\tilde{C}}_{n}^{b_{B}}{\tilde{C}}_{b_{A}}^{n}={\tilde{C}}_{b_{A}}^{b_{B}}=\left[\begin{array}{ccc} {\tilde{c}}_{11AB} & {\tilde{c}}_{12AB} & {\tilde{c}}_{13AB}\\ {\tilde{c}}_{21AB} & {\tilde{c}}_{22AB} & {\tilde{c}}_{23AB}\\ {\tilde{c}}_{31AB} & {\tilde{c}}_{32AB} & {\tilde{c}}_{33AB} \end{array}\right].$$

Taking ${\tilde{c}}_{21AB}$ as the measurement, we then have
(25)$$\begin{array}{ll} {\tilde{c}}_{21AB} & =-{\tilde{c}}_{11A}\left[{\tilde{c}}_{22B}\left(\phi_{DA}-\phi_{DB}\right)-{\tilde{c}}_{32B}\left(\phi_{EA}-\phi_{EB}\right)\right]\\ & \;-{\tilde{c}}_{21A}\left[-{\tilde{c}}_{12B}\left(\phi_{DA}-\phi_{DB}\right)+{\tilde{c}}_{32B}\left(\phi_{NA}-\phi_{NB}\right)\right]\\ & \;-{\tilde{c}}_{31A}\left[{\tilde{c}}_{12B}\left(\phi_{EA}-\phi_{EB}\right)-{\tilde{c}}_{22B}\left(\phi_{NA}-\phi_{NB}\right)\right]+\Delta \varphi_{AB}\\ & =\left({\tilde{c}}_{21A}{\tilde{c}}_{12B}-{\tilde{c}}_{11A}{\tilde{c}}_{22B}\right)\left(\phi_{DA}-\phi_{DB}\right)+\left({\tilde{c}}_{11A}{\tilde{c}}_{32B}-{\tilde{c}}_{31A}{\tilde{c}}_{12B}\right)\left(\phi_{EA}-\phi_{EB}\right)+\\ & \;\left({\tilde{c}}_{31A}{\tilde{c}}_{22B}-{\tilde{c}}_{21A}{\tilde{c}}_{32B}\right)\left(\phi_{NA}-\phi_{NB}\right)+\Delta \varphi_{AB} \end{array}$$

Hence, the modified observation model is given by
(26)$$\begin{array}{ll} \bar{z}\left(t\right) & ={\left[\begin{array}{ccccc} \delta v_{NA} & \delta v_{EA} & \delta v_{NB} & \delta v_{EB} & {\tilde{c}}_{21AB} \end{array}\right]}^{T}\\ & =\left[\begin{array}{ccccc} 0_{2\times 3} & I_{2\times 2}^{} & 0_{2\times 3} & 0_{2\times 2} & 0_{2\times 4}\\ 0_{2\times 3} & 0_{2\times 2} & 0_{2\times 3} & I_{2\times 2}^{} & 0_{2\times 4}\\ H_{1\times 3}^{1} & 0_{1\times 2} & -H_{1\times 3}^{1} & 0_{1\times 2} & 0_{1\times 4} \end{array}\right]\bar{X}\left(t\right)+\left[\begin{array}{c} \upsilon_{1}\left(t\right)\\ \Delta \varphi_{AB}\left(t\right) \end{array}\right]\\ & =\bar{H}\left(t\right)\bar{X}\left(t\right)+\bar{\upsilon }\left(t\right),\;\bar{\upsilon }\left(t\right)∼N\left(0,\bar{R}\right) \end{array}$$
where $H_{1\times 3}^{1}$ is expressed as
(27)$$H_{1\times 3}^{1}=\left[\begin{array}{ccc} {\tilde{c}}_{31A}{\tilde{c}}_{22B}-{\tilde{c}}_{21A}{\tilde{c}}_{32B} & {\tilde{c}}_{11A}{\tilde{c}}_{32B}-{\tilde{c}}_{31A}{\tilde{c}}_{12B} & {\tilde{c}}_{21A}{\tilde{c}}_{12B}-{\tilde{c}}_{11A}{\tilde{c}}_{22B} \end{array}\right].$$
The observation noise covariance matrix $\bar{R}$ takes the following form:(28)$$\bar{R}=\left[\begin{array}{ccc} R_{1} & 0_{2\times 2} & 0_{2\times 1}\\ 0_{2\times 2} & R_{1} & 0_{2\times 1}\\ 0_{1\times 2} & 0_{1\times 2} & r_{\varphi }^{2} \end{array}\right].$$

Consequently, the piecewise combined Kalman filter (PC-KF) based on the error dynamic model shown in Equation (18) and the modified observation model shown in Equation (26) can be designed. The corresponding two-position initial alignment algorithm with relative azimuth constraints is denoted as RATP in the following sections.

### 2.5. Comparison of Simulation Results among Different Alignment Schemes

According to the system dynamic error model constructed in Section 2, the third diagonal element of the covariance matrix P(t) in the conventional TP and ARTP algorithms, which is denoted as $P_{\left(3,3\right)}\left(t\right)$, or the eighth element diagonal of the covariance matrix P(t) in the new RATP algorithm, which is denoted as $P_{\left(8,8\right)}\left(t\right)$, may be regarded as a quantitative measurement of observability of the azimuth error and the alignment performance [23,26,29]. This section uses covariance simulation to compare $P_{\left(3,3\right)}\left(t\right)$ or $P_{\left(8,8\right)}\left(t\right)$ among the three alignment schemes. The simulation parameters are set in Table 1.

Simulation analysis is divided into two parts. The first part analyzes the variation of 1−σ value of the final estimated azimuth error with the total alignment time changing from 30 to 300 s under different angular vibration conditions. The simulation results can be seen in Table 2 and from Figure 1, Figure 2 and Figure 3. It is shown that the 1−σ final azimuth error of RATP is smaller than that of TP and ARTP. The alignment accuracy of RATP is less affected by angular vibration conditions compared with that of ARTP under different angular vibration conditions. The estimated azimuth error of ARTP tends to be equal to TP if the alignment time is long enough. The performance of ARTP is better than that of TP under stationary conditions, but its performance becomes worse when there are angular vibrations. In the case of angular vibration, ARTP can achieve faster convergence rate than TP for rapid alignment, but its accuracy is inferior to TP when the alignment time is longer. The second part analyzes the azimuth error convergence process through time history curves of the three schemes with the same total alignment time (180 s). The simulation result can be seen in Figure 1, Figure 2 and Figure 3. The convergence speed of RATP is faster than that of TP and ARTP, and ARTP is even slower than TP under large angular vibration conditions. Therefore, RATP is more suitable for rapid initial alignment.

Con.1 represents the stationary condition, Con.2 represents angular vibrations with an amplitude of about 5 arcsec and a random frequency of 5–10 Hz, and Con.3 represents angular vibrations with an amplitude of about 10 arcsec and a random frequency of 5–10 Hz.

## 3. Results and Discussion

### Two-Position Initial Alignment Experiment

Practical experimental tests were carried out to compare the two-position alignment performance by using the three schemes described earlier. The total alignment time is 180 s. For the first 90 s, the IMU is in state A, then the IMU rotated $180^{\circ }$ to state B. The experimental environment is shown in Figure 4. The main parameters of the inertial devices are shown in Table 3.

The velocity error curves under different conditions are shown in Figure 5. We can see from Figure 5 that the velocity errors of the three schemes are all less than 0.0005 m/s under different angular vibrations. Therefore, the velocity errors of the three schemes all meet the performance requirements. Figure 6 shows the horizontal curves of the three schemes, hence, we can make the conclusion that TP, ARTP, and RATP have a rate of convergence of horizontal attitude error which is similar with the velocity error. Furthermore, the convergence time of horizontal attitude errors is less than 10 s when SINS is in state B.

The azimuth errors of the three schemes for 180 s rapid alignment are shown in Table 4, and Figure 7, Figure 8 and Figure 9. The 1−σ values of the final azimuth error of TP at 180 s are 89.26, 117.23, and 141.83 arcsec in the different experimental conditions. The 1−σ values of the final azimuth error of ARTP at 180 s are 61.96, 124.01, and 254.55 arcsec, respectively. The convergence rate of RATP is better than that of TP under static conditions, and its performance decreases obviously compared with TP when angular vibration increases. As shown in Table 4 and Figure 8, the 1−σ values of final azimuth error of RATP at 180 s are 43.75, 50.75, and 87.83 arcsec, respectively. RATP shows superior performance under different conditions. Above all, the azimuth error convergence to a steady value is much faster for RATP than for the other two schemes under different conditions. The azimuth error accuracy of RATP in the same total alignment time is significantly higher than that of the traditional algorithms. Therefore, the experimental results have shown the effectiveness and superiority of the proposed algorithm.

## 4. Conclusions

To improve alignment precision and reduce the time needed for error convergence, this paper proposes a novel piecewise combined Kalman filter for state estimation. A two-position ground alignment algorithm for SINS is designed based on the proposed piecewise combined Kalman filter. Simulation results show that the proposed RATP algorithm presents better performance under different experimental conditions compared with the conventional alignment algorithms. ARTP performs better than TP under stationary condition, and it can achieve a faster convergence rate than TP for rapid alignment, but the accuracy of ARTP accuracy was inferior to TP when the alignment time under angular vibrations was longer. The real experimental results indicate that for the SINS in this paper, the azimuth error of RATP is about 40% less than that of the conventional algorithms under stationary or angular vibration conditions for 180 s rapid alignment. The proposed two-position ground alignment scheme, which is based on a piecewise combined Kalman filter and azimuth constraint information, has important engineering value for rapid SINS self-alignment.

## Figures and Tables

**Figure 1 sensors-19-01125-f001:**
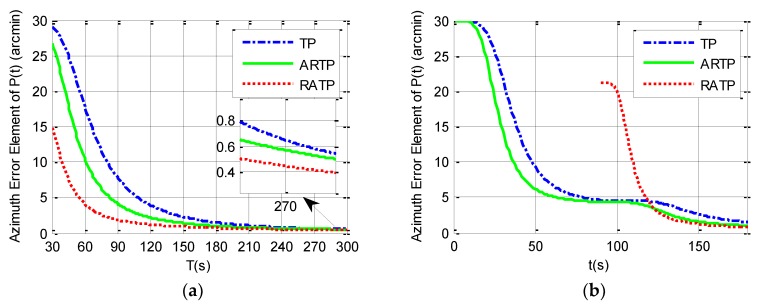
Simulation results under stationary condition. (**a**) 1−σ final azimuth error when the total alignment time *T* changes from 30 to 300 s. (**b**) Azimuth error time history curve when the total alignment time is 180 s.

**Figure 2 sensors-19-01125-f002:**
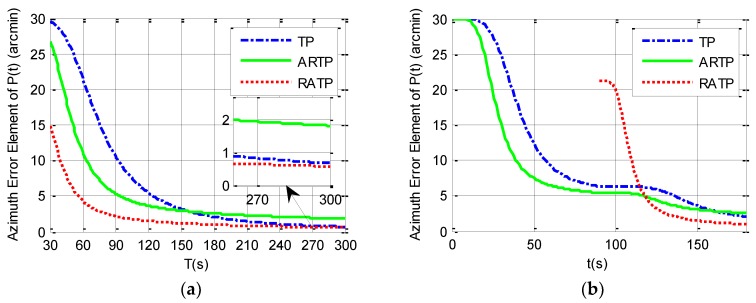
Simulation results under angular vibrations with an amplitude of 5 arcsec and a random frequency between 5 and 10 Hz. (**a**) 1−σ final azimuth error when the total alignment time *T* changes from 30 to 300 s. (**b**) Azimuth error curve when the total alignment time is 180 s.

**Figure 3 sensors-19-01125-f003:**
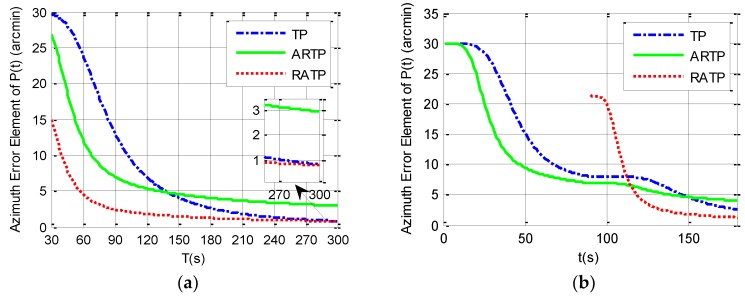
Simulation results under angular vibrations with an amplitude of 10 arcsec and a random frequency between 5 and 10 Hz. (**a**) 1−σ final azimuth error when the total alignment time *T* changes from 30 to 300 s. (**b**) Azimuth error curve when the total alignment time is 180 s.

**Figure 4 sensors-19-01125-f004:**
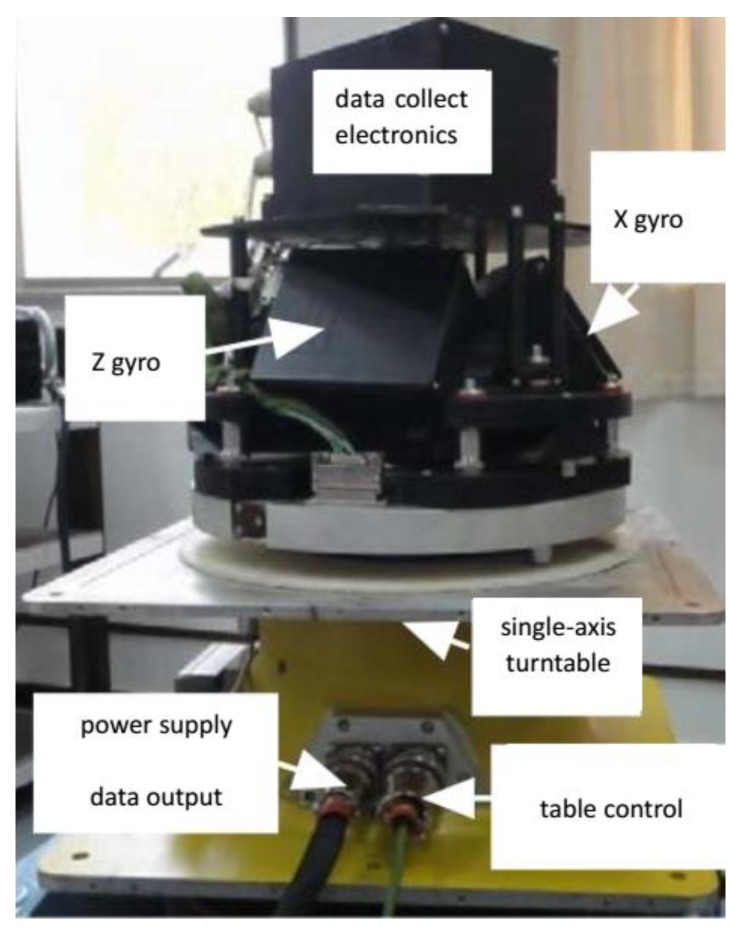
The experimental environment.

**Figure 5 sensors-19-01125-f005:**
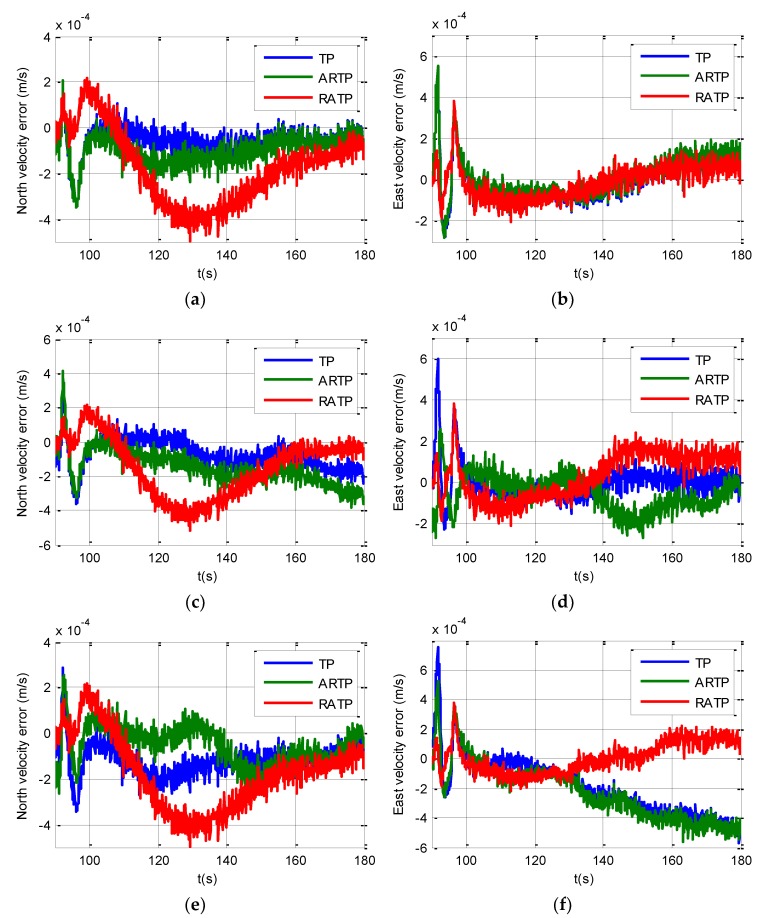
Velocity curves under different conditions from 90 to 180 s. (**a**) North velocity error under stationary condition; (**b**) East velocity error under stationary condition; (**c**) North velocity error under angular vibrations with an amplitude of about 5 arcsec and a random frequency between 5 and 10 Hz; (**d**) East velocity error under angular vibrations with an amplitude of about 5 arcsec and a random frequency between 5 and 10 Hz; (**e**) North velocity error under angular vibrations with an amplitude of about 10 arcsec and a random frequency between 5 and 10 Hz; (**f**) East velocity error under angular vibrations with an amplitude of about 10 arcsec and a random frequency between 5 and 10 Hz.

**Figure 6 sensors-19-01125-f006:**
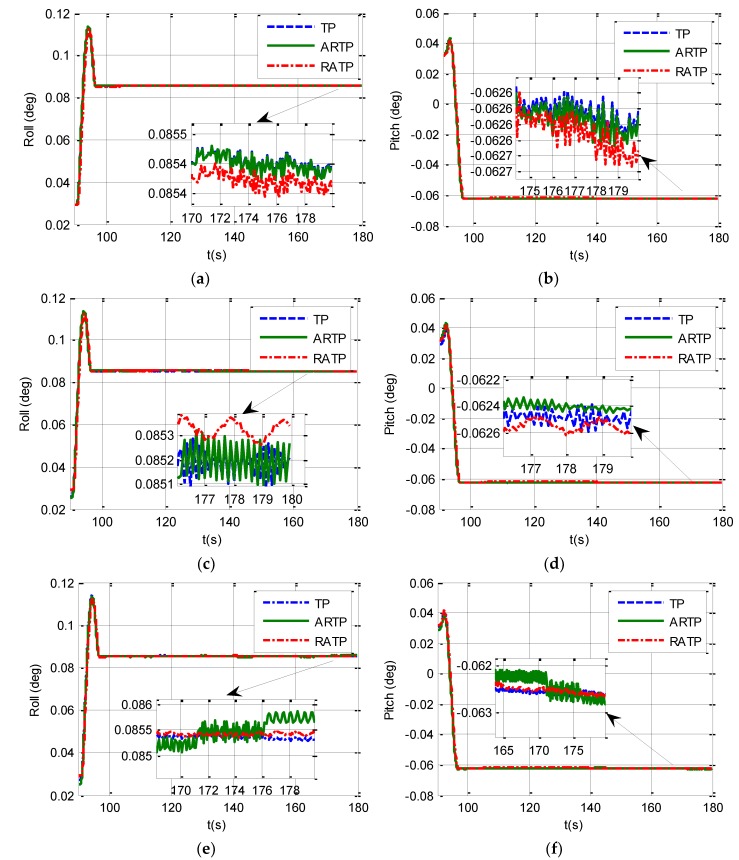
Horizontal curves under different conditions from 90 to 180 s. (**a**) Roll curve under stationary condition; (**b**) Pitch curve under stationary condition; (**c**) Roll curve under angular vibrations with an amplitude of about 5 arcsec and a random frequency between 5 and 10 Hz; (**d**) Pitch curve under angular vibrations with an amplitude of about 5 arcsec and a random frequency between 5 and 10 Hz; (**e**) Roll curve under angular vibrations with an amplitude of about 10 arcsec and a random frequency between 5 and 10 Hz; (**f**) Pitch curve under angular vibrations with an amplitude of about 10 arcsec and a random frequency between 5 and 10 Hz.

**Figure 7 sensors-19-01125-f007:**
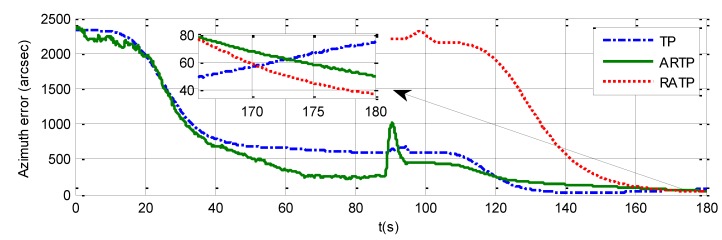
Azimuth error time history curve under stationary conditions when the total alignment time is 180 s.

**Figure 8 sensors-19-01125-f008:**
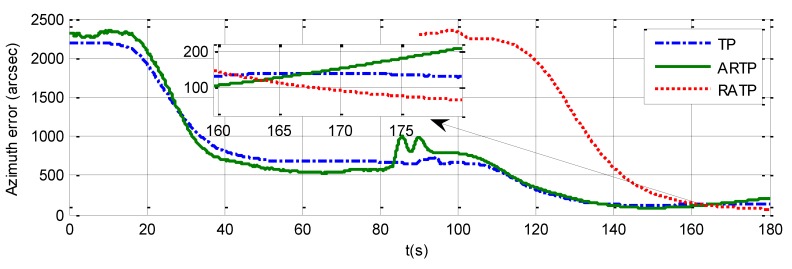
Azimuth error time history curve under angular vibrations with an amplitude of about 5 arcsec and a random frequency between 5 and 10 Hz when the total alignment time is 180 s.

**Figure 9 sensors-19-01125-f009:**
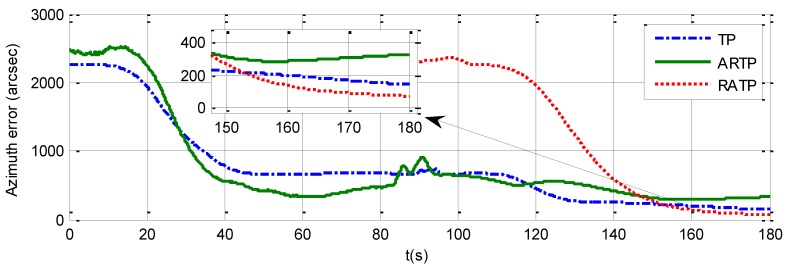
Azimuth error time history curve angular vibrations with an amplitude of about 10 arcsec and a random frequency between 5 and 10 Hz when the total alignment time is 180 s.

**Table 1 sensors-19-01125-t001:** Parameters for simulation.

Local Latitude L	28.21°
Bias instability of gyros Constant bias of gyros	$0.003^{\circ }/h$ $0.015^{\circ }/h$
Angular random walk of gyros	$0.0005^{\circ }/\sqrt{h}$
Bias instability of accelerometers Constant bias of accelerometers	20 µg 100 µg
Noise power spectrum density of accelerometers	$20\mu g/\sqrt{Hz}$
Process noise covariance parameters in $Q,\bar{Q}$	$n_{a}=20\;\mu g/\sqrt{Hz}$, $n_{g}=0.0005^{\circ }/\sqrt{h}$
Measurement noise covariance parameters of **TP** in $R_{1}$	$r_{v}=0.01\;m/s$
Measurement noise covariance parameters of **ARTP** in $R_{2}$	$r_{v}=0.01\;m/s$, $r_{g}=0.01^{\circ }/s$ (Con.1), $r_{g}=0.1^{\circ }/s$ (Con.2), $r_{g}=0.2^{\circ }/s$ (Con.3)
Measurement noise covariance parameters of **RATP** in $\bar{R}$	$r_{v}=0.01\;m/s$, $r_{\varphi }=0.012^{\circ }$ (Con.1), $r_{\varphi }=0.018^{\circ }$ (Con.2) $r_{\varphi }=0.024^{\circ }$ (Con.3)
Initial error covariance parameters of **TP** and **ARTP** in $P_{10}^{},P_{20}^{}$	$P_{10}^{}=P_{20}^{}=diag\left(\begin{array}{l} {\left(0.1^{\circ }\right)}^{2},{\left(0.1^{\circ }\right)}^{2},\;{\left(0.5^{\circ }\right)}^{2},{\left(0.01\;m/s\right)}^{2},{\left(0.01\;m/s\right)}^{2},\\ {\left(0.015^{\circ }/h\right)}^{2},{\left(0.015^{\circ }/h\right)}^{2},{\left(100\;\mu g\right)}^{2},{\left(100\;\mu g\right)}^{2} \end{array}\right)$
Initial error covariance parameters of **RATP** in $P_{30}^{}$	$P_{30}^{}=diag\left(\begin{array}{l} {\left(0.1^{\circ }\right)}^{2},{\left(0.1^{\circ }\right)}^{2},\;{\left(0.5^{\circ }\right)}^{2},{\left(0.01\;m/s\right)}^{2},{\left(0.01\;m/s\right)}^{2},\\ {\left(0.1^{\circ }\right)}^{2},{\left(0.1^{\circ }\right)}^{2},\;{\left(0.5^{\circ }\right)}^{2},{\left(0.01\;m/s\right)}^{2},{\left(0.01\;m/s\right)}^{2},\\ {\left(0.015^{\circ }/h\right)}^{2},{\left(0.015^{\circ }/h\right)}^{2},{\left(100\;\mu g\right)}^{2},{\left(100\;\mu g\right)}^{2} \end{array}\right)$

Con.1 represents the stationary condition, Con.2 represents angular vibrations with an amplitude of approximately 5 arcsec and a random frequency of 5–10 Hz, and Con.3 represents angular vibrations with an amplitude of approximately 10 arcsec and a random frequency of 5–10 Hz. TP conventional two-position initial alignment algorithm; ARTP, angular rate measurement-augmented two-position initial alignment algorithm; RATP, two-position initial alignment algorithm with relative azimuth constraints.

**Table 2 sensors-19-01125-t002:** Performance comparison of three alignment schemes.

Condition	1−σ Final Azimuth Error Value in 180 s (arcmin)			1−σ Final Azimuth Error Value in 300 s (arcmin)		
	TP	ARTP	RATP	TP	ARTP	RATP
**Con.1**	1.51	1.01	0.72	0.54	0.50	0.40
**Con.2**	2.06	2.55	0.96	0.67	1.81	0.58
**Con.3**	2.61	4.01	1.21	0.80	2.93	0.78

**Table 3 sensors-19-01125-t003:** Main parameters of inertial devices.

**Laser gyro**	Zero-bias stability	$0.003^{\circ }/h(1\sigma )$
	Angle random walk	$0.0005^{\circ }/\sqrt{h}\left(1\sigma \right)$
**Quartz pendulous accelerometer**	Zero-bias stability	20 μg(1σ)
	Noise power spectrum density	$20\;\mu g/\sqrt{Hz}\left(1\sigma \right)$
	Sampling frequency	500 Hz

**Table 4 sensors-19-01125-t004:** Azimuth error under different conditions for 180 s rapid alignment.

Num	TP (arcsec)			ARTP (arcsec)			RATP (arcsec)		
	Con.1	Con.2	Con.3	Con.1	Con.2	Con.3	Con.1	Con.2	Con.3
**1**	−110.26	−199.00	272.94	−95.08	−52.74	−94.134	−1.76	45.31	144.47
**2**	−38.26	56.85	−183.61	9.02	−148.55	51.26	34.64	10.83	−69.82
**3**	110.46	−24.70	34.60	7.41	78.65	59.75	3.39	−58.99	14.40
**4**	−70.55	237.23	68.07	−59.90	−334.08	−66.70	−40.21	−53.06	90.46
**5**	6.14	−73.86	−254.07	35.43	217.33	−132.19	−59.04	12.49	−127.33
**6**	102.48	3.48	62.07	103.11	239.40	182.01	62.99	43.42	−52.17
**7**	−96.24	−18.10	−124.30	−4.95	−7.66	−476.15	−8.63	−20.32	−50.61
**8**	−84.72	−91.60	−20.84	−85.77	−157.55	−14.43	−78.16	−85.72	65.65
**9**	18.52	100.76	−39.40	29.03	−83.69	−191.20	28.56	37.15	−107.95
**10**	104.79	130.65	112.16	81.13	101.39	508.14	44.86	77.35	85.82
**11**	−55.33	−97.28	91.85	−45.65	−54.37	−142.47	−22.21	−40.75	−50.62
**12**	124.49	−24.43	−19.46	26.22	201.87	−316.57	35.58	32.29	57.71
**STD value**	**89.26**	**117.23**	**141.83**	**61.96**	**174.01**	**254.55**	**43.75**	**50.75**	**87.83**

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
