# Peer review of "Rapid SINS Two-Position Ground Alignment Scheme Based on Piecewise Combined Kalman Filter and Azimuth Constraint Information"

_sensors, 2019, doi:10.3390/s19051125_

Round 1

Reviewer 1 Report

Section 2 presents the the core discriminator of this article, specifically, the development of the "Piecewise Combined Kalman Filter," however, the underlying system model development in section 2.1 does not cite any references and is pivotal to the subsequent sub-sections.  Please provide citations therein.

Author Response

Dear Editor,

I have modified the manuscript according to the reviews of experts. Please forward the modified manuscript to the experts.

Kind regards,

Lu Zhang

Reviewer 2 Report

Dear Authors,

the paper is interesting and the topic, you have written is actual I have no some serious recommendations, but the paper can be a little bit improved.

- the introduction section can be extended about some new methods of alignment.

- The experimental results should be better described, is written, experimental results, so I expected, you will describe better the measurement setup, some photos, etc. Also, the results should be discussed better.

Best regards,

Reviewer

Author Response

(The authors gave the same response as above.)
